# Infant Exposure to Antituberculosis Drugs via Breast Milk and Assessment of Potential Adverse Effects in Breastfed Infants: Critical Review of Data

**DOI:** 10.3390/pharmaceutics15041228

**Published:** 2023-04-13

**Authors:** Engi Abdelhady Algharably, Reinhold Kreutz, Ursula Gundert-Remy

**Affiliations:** Charité–Universitätsmedizin Berlin, Corporate Member of Freie Universität Berlin and Humboldt-Universität zu Berlin, Institute of Clinical Pharmacology and Toxicology, Charitéplatz 1, 10117 Berlin, Germany; engi.algharably@charite.de (E.A.A.);

**Keywords:** tuberculosis, human milk, breastfeeding, drug-exposed infants, adverse drug effects

## Abstract

Infants of mothers treated for tuberculosis might be exposed to drugs via breast milk. The existing information on the exposure of breastfed infants lacks a critical review of the published data. We aimed to evaluate the quality of the existing data on antituberculosis (anti-TB) drug concentrations in the plasma and milk as a methodologically sound basis for the potential risk of breastfeeding under therapy. We performed a systematic search in PubMed for bedaquiline, clofazimine, cycloserine/terizidone, levofloxacin, linezolid, pretomanid/pa824, pyrazinamide, streptomycin, ethambutol, rifampicin and isoniazid, supplemented with update references found in LactMed^®^. We calculated the external infant exposure (EID) for each drug and compared it with the recommended WHO dose for infants (relative external infant dose) and assessed their potential to elicit adverse effects in the breastfed infant. Breast milk concentration data were mainly not satisfactory to properly estimate the EID. Most of the studies suffer from limitations in the sample collection, quantity, timing and study design. Infant plasma concentrations are extremely scarce and very little data exist documenting the clinical outcome in exposed infants. Concerns for potential adverse effects in breastfed infants could be ruled out for bedaquiline, cycloserine/terizidone, linezolid and pyrazinamide. Adequate studies should be performed covering the scenario in treated mothers, breast milk and infants.

## 1. Introduction

Tuberculosis (TB) is a major global public health problem estimated to affect around 6.4 million people worldwide, according to the latest WHO report in 2021 [1]. A global incidence rate of 3.6% was reported in 2021, with a third of the cases affecting women, especially in the reproductive age [1]. Maternal infection with TB during pregnancy, delivery and postpartum has been associated with unfavorable outcomes for both pregnant women and infants. The actual global burden of pregnant women with active TB is not currently known; hence, an estimation of the prevalence is given in the literature depending on the WHO region, which ranges from 0.06% to 0.25% in low-burden countries to 0.07% and 0.5% in high-burden countries among HIV-negative women [2]. This is also supported by individual reports on the prevalence emerging from developed countries [3] and developing/low-income countries [4]. A recent study in Mozambique [5] reported a TB prevalence of 0.5% and 0.3% among pregnant and postpartum women, respectively. Another study in Benin reported a prevalence of 0.049% [6].

Breastfeeding women are an important sub-population of TB patients, pertaining to their continuous treatment and opting to breastfeed their infants given the fact that the mother’s milk is the most appropriate food for the infant containing all the necessary components of food [7].

According to the latest WHO guidelines from 2022 [8], drug-susceptible TB is treated with a 6-month regimen of isoniazid, rifampicin, ethambutol and pyrazinamide. As for drug-resistant TB, regimens are now recommended which include bedaquiline, pretomanid, linezolid and levofloxacin or moxifloxacin [8].

Most guidelines encourage breastfeeding in women with drug-susceptible TB who are deemed non-infectious [9] since breast milk does not contain the tubercle bacilli which renders TB transmission to the infant via the milk unlikely [10]. Since drugs taken by the lactating mother can be passed from the maternal circulation to the mother’s milk and consequently to the breastfed infant, there is a valid concern regarding the extent of transfer and the effects of the drugs on breastfed infants. Although most of the antituberculosis (anti-TB) drugs used in drug-susceptible TB are excreted into breast milk, treatment with these drugs is said to be safe for the infant [9]. However, some uncertainties regarding the safety of exposure to anti-TB agents exist and the extent to which these drugs are excreted in the breast milk of lactating mothers, especially for newer agents, remains largely unresolved.

Many of the anti-TB agents are eliminated from the body through metabolism in the liver. In breastfed infants, accumulation may occur and might lead to increasing blood concentrations and internal exposure than in an adult due to the low expression of metabolizing enzymes, especially in premature infants [10,11,12]. Thus, the relationship between external exposure and internal exposure in infants is different from that in their mothers. Hence, in infants, the higher internal exposure might be related to a higher risk for adverse effects.

The amount of drug excreted into human milk depends on a number of factors (Figure 1). Drug-related factors determining a drug’s partitioning into the milk include physicochemical properties such as the acid base dissociation constant (pKa), molecular weight, as well as lipid solubility. Lipid soluble drugs tend to be concentrated in milk since milk contains more lipid than plasma whereas compounds with a higher molecular weight are less likely to enter the milk compartment. Drugs enter the milk primarily by passive diffusion depending on the equilibrium achieved between the maternal plasma and the milk compartment. The maternal drug plasma concentration, determined by the dosing regimen and the drug clearance is the driving force for drug diffusion [13].

The amount of drug the infant is exposed to during the breastfeeding period can be estimated by measuring the drug’s concentration in the milk and the average volume of the daily consumed milk by the infant which gives the “external” infant dose (EID). The “internal” dose, the systemic dose in the body of the infant, depends on the gestational development. Immaturity of metabolism and renal excretion in preterm infants and newborns within the first three to six months leads to a higher internal exposure of the same external exposure compared to older infants and adults [11,12,13,14]. Physiologically based pharmacokinetic (PBPK) modelling is a non-invasive approach to gain information on the internal exposure of the infant by simulating the concentration–time profile when the nursing mother is treated with therapeutic doses of anti-TB drugs. The model typically consists of a nursing woman model treated by the drug and its excretion into the milk, connected to a second model for the breastfed infant that consumes the milk and is exposed to the drug by this route. The age-specific physiological data and the immature metabolizing and excretory function in the breastfed infant in the first three to six months are captured in the model (Figure 2).

The existing information on the exposure of breastfed infants lacks a systematic approach and a critical review of the published data. The aim of this work is to retrieve and to evaluate the quality of the existing information regarding the concentrations of anti-TB drugs in the plasma and milk and, if available, in the plasma of infants in a systematic manner, scaling the concentrations to the standard doses recommended for the individual drug. Adverse effects of the drugs, as given in authoritative sources, will be screened for dose-dependency and the external infant doses and, if available, information on the internal doses will be assessed for dose-dependent adverse effects.

Thus, the presented information will be better structured and the basis for recommendations regarding the administration of anti-TB drugs during lactation in treated mothers will be made more transparent. From our findings, we will develop some recommendations regarding how, in the future, more valid information could be obtained.

## 2. Methods

From the WHO list of anti-TB drugs [8], we selected isoniazid, rifampicin, ethambutol, pyrazinamide, streptomycin, levofloxacin, bedaquiline, linezolid, clofazimine, cycloserine, terizidone, pretomanid/PA824.

A systematic literature search was performed in PubMed (accessed on 20 January 2023) for each anti-TB drug using the search strings pharmacokinetics [MeSH term]; plasma concentration [title/abstract] OR plasma concentrations [title/abstract] and the name of the anti-TB drug (pyrazinamide, streptomycin, levofloxacin, bedaquiline, linezolid, clofazimine, cycloserine, terizidone, pretomanid/PA824) without time constraints. A second search was performed for collecting data on the anti-TB drugs in human milk using the string human milk concentration [MeSH term] and the name of the anti-TB drug (pyrazinamide, streptomycin, levofloxacin, bedaquiline, linezolid, clofazimine, cycloserine, terizidone, pretomanid/PA824) without time constraints. A third search was performed for collecting data on the anti-TB drugs in breastfed infants using the string infant plasma concentration [MeSH term] and the name of the anti-TB drug (pyrazinamide, streptomycin, levofloxacin, bedaquiline, linezolid, clofazimine, cycloserine, terizidone, pretomanid/PA824) without time constraints (Appendix A).

Time constraints (December 2017 until December 2022) were used for rifampicin, ethambutol and isoniazid, as the data until 2017 were collected, evaluated and used in our publications [15,16].

Furthermore, we checked the LactMed^®^ database to identify additional, potentially relevant, records which were not retrieved by the literature search. New records were then added to supplement the retrieved articles from PubMed searching.

The publications found were screened for relevance by title and abstract and in a second step by full-text screening. Exclusion criteria for not being relevant were in vitro studies, in vivo studies in animals, description of an analytical method, bioavailability studies, studies in healthy subjects, no relevant data (see below) reported in the publication. One publication was excluded because it was in the Russian language and could not be retrieved in full text.

Relevant data were extracted (plasma concentrations, area under the concentration-time profile (AUC), concentrations in human milk, concentrations in the breastfed infant) whereby, concerning plasma concentrations/AUC, those in treated nursing women had priority. If they were not available, the next priority was data from patient population kinetic analysis. If no population kinetic data from treated patients were available, data from single studies on patients were evaluated. If data from patients were not available, population kinetic data were otherwise obtained, and if population kinetics were not available, data from single studies were evaluated. The quality of the selected kinetic studies was evaluated using a checklist developed by Gafar et al. [17] who performed a systematic review of anti-TB drug pharmacokinetics (PK) in children and adolescents (Appendix A). Gafar et al. [17] used a questionnaire of 20 questions related to the information in the publications on the purpose of the study, selection of study subjects, timeline of sampling, sampling and storage conditions, analytical, PK and statistical procedures, attrition rate and presentation of results. The answers were: yes, no or not applicable. The maximal score which could be attained was 20, i.e., 20/20. If some questions were to be answered “not applicable”, the maximum which could be attained by “yes” answers was reduced (e.g., 18/18). The quality of the available data from PK studies was thus described according to the calculated score as either “satisfactory” or “not satisfactory”.

We selected the AUC divided by the time, reflecting the average concentration as the most relevant kinetic information. If these data were not reported in the publication, we performed the calculations based on the data for the AUC and time span given in the publication.

From the data on the average plasma concentrations, the plasma concentration was selected which corresponded to the concentration resulting from a standard dose; if another dose was given in the publication, we scaled the data to the standard dose assuming linear kinetics.

If, in the same study, both plasma concentration and milk concentration were reported, the ratio between milk concentration and plasma concentration was calculated. If human milk concentrations were reported without concomitant plasma concentrations, we calculated an average milk concentration adjusted to the standard dose; in a second step, the ratio between milk concentration and plasma concentration selected from the information found, adjusted to the standard dose, was calculated. The quality of the available data was evaluated using several factors such as the plausibility of the provided milk data in relation to the dose given to the mother, the number of subjects, the number of samples (concentration data points) and the timing of sample collection in the studies; hence, data were qualified as “satisfactory” or “not satisfactory.

Figure 3 shows the flowchart of the procedure.

For pretomanid, no milk concentration data were available and we estimated the concentration in human milk using the equations for calculating milk to plasma ratio given in Abduljalil et al. [18] (Appendix A). The relevant parameters were taken from different sources (Appendix A).

The concentration in the milk ($C_{m}$) used for estimating the external infant dose was calculated using the formula:(1)$$C_{m}=\frac{M}{p}\times C_{\mathrm{average}}$$
where $C_{\mathrm{average}}$ is the average maternal plasma concentration.

The daily external exposure of the infant (EID) to the anti-TB drug was calculated by multiplying the milk concentration by the daily milk consumption of the infant.
(2)$$\mathrm{EID}=C_{m\;}\times 0.185$$
where 0.185 L/kg bw per day is the average of the amount of milk consumed per day by exclusively breastfed infants between 8 days and 4 months according to the EFSA guidance on the risk assessment of substances present in food intended for infants below 16 weeks of age [19].

The relative external infant dose (REID) was calculated by comparing the calculated daily external dose with that for the treated infant according to dose recommendations in WHO guidelines [20].
(3)$$\mathrm{REID}\;=\frac{{\mathrm{EID}}_{\mathrm{day}}}{{\mathrm{Infant}\;\mathrm{dose}}_{\mathrm{day}}\;}$$

For rifampicin, ethambutol and isoniazid, three publications were available where external exposure as well as internal exposure in breastfed infants were modelled by PBPK modelling. In the model, the output via breast milk of the nursing mother was used as input to the breastfed infant. Age-specific anatomical and physiological data were used to parameterize the nursing mother and the breastfed infant with particular emphasis on the immature metabolism of the infants. For ethambutol, for example, the clearance was only 35% of the clearance in the adult, indicating that a comparison of the external doses between nursing mothers and breastfed infants could be misleading [16].

To enable an assessment of whether exposure to the external or, if available, internal exposure in the infant would raise health concerns, we consulted official sources such as approved material by a regulatory body (i.e., the official summary of product characteristics; SPC) and published material by a UK-based learned society produced by Potter et al. [21], updated in 2021. The adverse effects enumerated there were assessed for whether they would be dose-dependent and whether a cut-off dose is known. Accordingly, we assessed the potential health impact of the infant dose.

## 3. Results

For the plasma concentration data, a total of 730 articles were retrieved, 678 were excluded after title and abstract screening, 52 were evaluated in full-text screening and after exclusion, 17 articles remained. For the milk concentration, a total of 23 articles were retrieved, 14 were excluded after title and abstract screening, 9 were evaluated in full-text screening and 7 articles remained. For the infant plasma concentration, a total of 74 articles were found, 70 were evaluated in full-text screening and after exclusion, 4 articles remained. In addition, three concentration–time profiles were available from PBPK modelling (Table 1; Appendix A). In the tables, the results are presented firstly, for anti-TB drugs for which PK studies including population kinetics are available followed by substances with PK and modelling studies in an alphabetical order.

### 3.1. Bedaquiline

#### 3.1.1. Kinetics

Several PK studies were found in the literature search (Table 1). A single, small, limited PK study was performed on nursing women reporting bedaquiline concentrations in human milk. Data points were also available in three breastfed infants, reflecting internal exposure [22]. Three population kinetic studies available were regarded as the primary source for information on the dose-related AUC and the calculated average concentration.

The study by Svensson et al. [23] was not considered further as the study did not report on the AUC or C_average_. It was rated 13/18, mainly because information on the patients was not sufficient and details on sampling were not provided.

The study by McLeay et al. [24] was also not taken into consideration as the AUC was not reported as a value but graphically displayed and further reported data did not allow its calculation. The study was of poor to medium quality (10/17).

The study by Kurasawa et al. [25] was taken into consideration even though the quality was limited (10/14). The study was performed on 16 healthy volunteers with an average AUC_0–72_ of 14.65 µg/mL × h and a calculated C_average_ of 0.2 µg/mL after a 100 mg single oral dose, corresponding to 1.54 mg/kg bw.

The small study by Court et al. [22] measured plasma concentrations in the mothers, and milk concentrations and plasma concentrations in the breastfed infants. The mothers were on a long-term therapeutic regimen with bedaquiline and were given the standard dose of 200 mg bedaquiline three times a week, corresponding to approximately 1.22 mg/kg per day. Measurements in five breastfeeding mothers were performed about six weeks antepartum. Seventy plasma samples from the five mothers and seven milk samples from two mothers were obtained. In addition, three plasma samples from three breastfed infants could be quantified. Quantification was performed with a reliable, validated method; the sampling and storage conditions were appropriate.

#### 3.1.2. M:P Ratio

The estimated M:P ratio was 13.6 for bedaquiline based on the one single data point reported by Court et al. [22].

#### 3.1.3. Infant Dose

The average bedaquiline concentration in the mothers’ PK samples obtained postpartum was 0.4 µg/mL. Using this value, the average C_m_ would be 6.7 µg/mL and, multiplying it by 0.185 L/kg bw as the daily infant milk consumption, the infant dose is 1.3 mg/kg bw per day. The REID result is 0.65 compared to the standard dose of 2 mg/kg per day in infant patients treated with bedaquiline, according to the WHO guidelines [20].

Notably, in two of the infants, the concentrations measured in the plasma were very low (about 0.03 µg/mL) and no milk concentration values were reported for their mothers. In one of the infants, the plasma concentration was about 0.3 µg/mL, nearly the same value as in the mother; the corresponding milk concentration in the mother was 6.7 µg/mL.

#### 3.1.4. Adverse Effects

The most relevant adverse effect is a prolongation of QTc which is presumed to be more common with hypokalemia, proarrhythmic conditions or when combined with other drugs that prolong the QT interval such as clofazimine, fluoroquinolones or macrolides. In two retrospective studies, 4.3% of 420 patients [26] and 15.2% of 104 patients [27] had a clinically significant QTc prolongation, indicating that this adverse effect is not rare.

The infant bedaquiline dose (1.3 mg/kg bw per day) is in the same range as the standard dose in infant patients treated with bedaquiline (2 mg/kg bw per day) [20].

Given the high exposure in breastfed infants and considering the statement of WHO on the lack of safety data on the use of bedaquiline in children <6 years [8], the potential of bedaquiline to elicit severe cardiac arrhythmias could raise concerns (Table 2).

### 3.2. Clofazimine

#### 3.2.1. Kinetics

One PK study was found (Table 1) which was performed on nursing women. Data were published on clofazimine concentrations in the plasma and human milk; however, no data were available in the breastfed infant to reflect internal exposure. The quality of the kinetic study [28] was rated as poor (5/18). The reliability of the applied analytical method, HPTLC, was unclear and no validation was described. The plasma concentrations of the mothers, taken 4-6 h following the intake of either 50 mg daily (n = 5), 100 mg every other day (n = 2) or 100 mg daily (n = 1) for an average of 5 months, were reported as 0.9 ± 0.03 µg/mL (range 0.8–1.0 µg/mL). The corresponding milk concentrations were 1.3 ± 0.09 µg/mL (range 0.8–1.7 µg/mL).

#### 3.2.2. M:P Ratio

The M:P is given as 1.48 ± 0.08 (range 1.0 to 1.7) [28].

#### 3.2.3. Infant Dose

The EID was calculated as 0.24 mg/kg bw per day, with a range of 0.15 to 0.32 mg/kg bw per day. No data were available in the breastfed infant to reflect internal exposure after long-term treatment of the mother with either 50 mg daily, 100 mg every other day or 100 mg daily (corresponding to 0.83 mg/kg or 1.67 mg/bw per day). The REID result was 0.05–0.12, considering the standard dose (2–5 mg/kg per day) [20] in infant patients treated with clofazimine.

#### 3.2.4. Adverse Effects

Common adverse effects reported in more than 50% of the patients encompass pink to brownish-black skin discoloration resembling sun-tanning (reversible after treatment discontinuation), gastrointestinal symptoms and weight loss. Risk of QT prolongation and ventricular tachyarrhythmias have been highlighted in case reports. Quantifying the effect of the clofazimine concentration on QT prolongation was performed in a recently published study by Abdelwahab et al. [29]. The QT effect was statistically and clinically significant. None of the participants in the clofazimine monotherapy arm dosed with a loading dose of 300 mg for 3 days, followed by 100 mg daily, crossed the threshold for life-threatening cardiac arrhythmias. However, considering that clofazimine is usually included in combination treatment regimens containing drugs such as bedaquiline, delamanid and fluoroquinolones, which prolong QT intervals, a risk could not be ruled out. Additive QT prolongation has been shown when clofazimine is combined with bedaquiline [30].

Skin discoloration in breastfed infants was reported in some cases. No reports have been published on the occurrence of gastrointestinal symptoms or QT prolongation. The REID of 0.05-0.12 compared to the standard therapeutic dose in infants indicates that the infant internal exposure seems not to raise health concerns (Table 2).

### 3.3. Cycloserine/Terizidone

#### 3.3.1. Kinetics

For cycloserine, several kinetic studies were found in the literature search (Table 1); however, none were performed on nursing women. Data were found on cycloserine concentrations in human milk, but the original publications were not accessible in full text. No data were available in the breastfed infants to reflect internal exposure.

According to the methods section, the available population kinetic studies were seen as the primary source for information on the dose-related AUC and the calculated C_average_. The quality of the study of Alghamdi et al. [31] was rated as poor to medium (7/16 points), mainly because information on the patients from 3 cumulatively analyzed studies were not sufficient, the analytical method was not described and the AUC for the patients was not established. Therefore, the study was not used as a primary source.

The study of van der Galiën et al. [32] was performed on patients. The quality of the study was rated fair to good (13/18). The reported AUC was 888 µg/mL × h (718–1252 µg/mL × h) (median; 25th–75th). The C_average_ resulted in 37 µg/mL with a dosing of 11.2 (9.75–12.8) mg/kg bw.

Three additional PK studies in healthy volunteers had very good quality [33,34,35]. Zhu et al. [35] rated 16/17, Zhou et al. [34] 16/17 and Park et al. [33] 16/18 and their results were included in our analysis.

Zhu et al. [35] reported a median AUC_0–48 h_ of 214 µg/mL × h (range 163–352 µg/mL × h) and a median C_average_ of 4.46 µg/mL (range 3.4–7.3 µg/mL) for a dose of 500 mg, corresponding to 8.3 mg/kg.

Zhou et al. [34] reported a mean AUC_0–72 h_ of 596.72 µg/mL × h ± 265.53 µg/mL × h and mean C_average_ of 8.3 µg/mL ± 3.7 µg/mL with a dose of 500 mg, corresponding to 8.9 mg/kg. Park et al. [33] reported an AUC_0–24_ at a steady state of 242.3 ± 99.8 µg/mL × h, thus resulting in a C_average_ of 10.1 ± 4.2 µg/mL for a dose of 250 mg b.i.d, corresponding to 7.5 mg/kg bw.

The differences between the three studies performed on healthy volunteers could be explained by the different study design (multiple versus single dose) and time span for the AUC determination. However, why the AUC and resulting C_average_ in the study of van der Galiën et al. [32] found in patients is four- to eightfold higher than the values in healthy subjects is difficult to explain.

In Tran and Montakantikul [36], a mean C_m_ of 13.4 µg/mL (range 6–19 µg/mL) after a 250 mg dose q.i.d was reported. However, the full text of the primary source [37] could not be retrieved. Similar values were provided in a table by Vorherr [38] without referencing the primary source for this information.

#### 3.3.2. M:P Ratio

Using the milk concentration of 13.4 µg/mL reported by Tran and Montakantikul [36] and the C_average_ at a steady state of Park et al. [33] corrected for dose, an M:P ratio of 0.66 was calculated.

#### 3.3.3. Infant Dose

The EID was calculated as 2.5 mg/kg bw per day which is nearly 20% of the standard therapeutic infant dose, according to the WHO guidelines [20]. No data were available in the breastfed infant to reflect internal exposure after a dose of 1000 mg daily in the mother, corresponding to 16.7 mg/kg bw per day.

#### 3.3.4. Adverse Effects

Neurological adverse effects such as confusion, disorientation, dizziness, somnolence and psychiatric side effects, such as depression, seizure and psychotic disturbances, are associated with an extremely high serum peak concentration above 35 µg/mL which could only occur with unusually high therapeutic doses. Congestive heart failure may develop in extremely rare cases if daily doses are greater than 1–1.5 g which is unusually high. In rare cases, megaloblastic anemia due to vitamin B12 and/or folic acid deficiency may develop.

Given the limitation of the analytical performance of the measurements obtained in the 1950s, limiting the validity of the reported milk concentration, the result of the calculated infant dose is fraught with high uncertainty. The REID result is 0.2 compared to the standard infant cycloserine dose (15–20 mg/kg per day) [20]. Although no adverse effects in breastfed infants have been reported in the literature, dose- and concentration-dependent adverse effects could not be ruled out (Table 2).

### 3.4. Levofloxacin

#### 3.4.1. Kinetics

Several PK studies were found in the literature search (Table 1); however, none were performed on nursing women. Data from one small study were found on levofloxacin concentrations in human milk. No data are available in the breastfed infant to reflect internal exposure. According to the methods section, the available seven population PK studies were considered as the primary source for information on the dose-related AUC and the calculated C_average_.

One of the studies [39] was not considered as it was performed on pediatric patients, thus not being suitable for representing adult patients.

Six population PK studies had to be excluded due to a lack of data for the AUC or C_average_ [40,41,42,43,44,45]. In the study of Zhang et al. [46], which was of fair to good quality (13/18), the plasma AUC was shown to be dependent on the creatinine clearance in 164 patients. The AUCs were 49.9 µg/mL × h and 56.1 µg/mL × h for a creatinine clearance of 120 mL/min and 100 mL/min, respectively. The C_average_ was calculated as 2.1 and 2.4 µg/mL (when scaled to a dose of 500 mg).

The study of van’t Boveneind-Vrubleuskaya [47] was of good quality (16/19). A total of 20 patients with a mean age of 31 years (interquartile [IQR] range: 27–35 years) were enrolled in this study and received 15 mg/kg levofloxacin once daily. The median AUC_0–24_ was 98.8 µg/mL × h (IQR 84.8–159.6 µg/mL × h). The C_average_ was calculated as 2.3 µg/mL (IQR 2.0–3.7 µg/mL) when scaled to a dose of 500 mg.

Mohamed et al. [48] published a study of 45 patients. Measurements were obtained in the plasma and with a separate analytic method in the saliva as a matrix which can be obtained by non-invasive sampling. The quality of the study rated fair to good (13/19). The median AUC_0–24_ in the serum was 140 (IQR:102.4–179.09) µg/mL × h and the median AUC_0–24_ in the saliva was 97.10 (IQR 74.80–121.10) µg/mL × h with a daily dose of 15 mg/kg, showing a positive linear correlation with the serum and saliva AUC_0–24_. The C_average_ was calculated as 3.2 µg/mL (IQR 2.0–3.7 µg/mL) when scaled to a dose of 500 mg.

Therefore, the results of the population PK studies of patients were in good conformity.

A further three conventional PK studies of healthy subjects were in excellent agreement regarding the resulting C_average_ calculated from the measured AUCs and all three studies were of fair to good quality; [49] (13/17), Chien, et al. [50] (14/17) and Lubasch et al. [51] (14/18). The calculated C_average_ were 1.9 ± 0.2 µg/mL [51], 2.0 ± 0.3 µg/mL [49] and 2.0 ± 0.4 µg/mL [50] when scaled to a dose of 500 mg.

Cahill et al. [52] measured levofloxacin milk concentrations in 1 woman treated with 500 mg levofloxacin for 23 days without concomitant measurements in the plasma. Random sampling was performed during treatment and the concentration–time profile was followed after discontinuation. The milk AUC in a 24 h dosing interval was approximately 120 μg/mL × h, from which an average C_m_ of 5 µg/mL was calculated.

#### 3.4.2. M:P Ratio:

The M:P ratio could not be calculated due to a lack of data.

#### 3.4.3. Infant Dose

For the infant exposure, the average C_m_ in the milk of 5 µg/mL, multiplying it by 0.185 L/kg bw was used, resulting in an EID of 0.925 mg/kg bw per day which is 9-fold lower than the adult daily dose of 500 mg (corresponding to 8.3 mg/kg bw per day) and 10- to 20-fold lower than the infant and young children dose of 15–20 mg/kg daily [20] with a REID of 0.06 and 0.05, respectively.

#### 3.4.4. Adverse Effects

The most relevant adverse effect is a prolongation of QTc which is said to be more common in hypokalemia, proarrhythmic conditions and in combination with other drugs that prolong the QT interval such as clofazimine, bedaquiline or macrolides. Torsade de pointes, related to QT prolongation, has been observed in a higher frequency associated with levofloxacin intake compared to amoxicillin intake [53].

Levofloxacin can trigger seizures by inhibiting the gamma-aminobutyric acid (GABA) binding to the GABA A receptor, especially in individuals with underlying central nervous system disease, and in the elderly with an impaired renal function. However, it is an uncommon adverse effect of levofloxacin [54].

Eosinophilia, thrombocytopenia and neutropenia are rare effects. Arthritis, tendon inflammation and rupture are well-known adverse effects which are, nevertheless, uncommon [55].

Given the 10- to 20-fold difference between the therapeutic dose in infants and the EID, there is no concern for adverse effects due to nursing (Table 2).

### 3.5. Linezolid

#### 3.5.1. Kinetics

Several PK studies were found in the literature (Table 1) with one study performed on one nursing woman. Data from three small studies were found regarding linezolid concentrations in human milk. One data point is available for the breastfed infant to reflect internal exposure.

The available seven population PK studies were seen as the primary source for the information on the dose-related AUC and the calculated C_average_.

One of the studies [56] was not considered as the study was performed on paediatric patients and so was not suitable for representing adult patients.

The study by Abe et al. [57] was not considered in our analysis since the study was of medium quality (14/20), mainly because information on the patients was not sufficient.

The study of Plock et al. [58] was not considered in our analysis as the study was of medium quality (15/20) as the AUC was not reported and further reported data did not allow for its calculation.

In addition, the study by Sakurai et al. [59] could not be considered further although the quality rating was very good (17/19). The AUC was presented only graphically after 10,000 Monte Carlo simulations and the further reported data did not allow for its calculation. The AUC, taken from the graph, had a median value of 500 mL/min × h (25 to 75 percentile 180–1000 mL/min × h) for an intravenous (i.v.) dose of 600 mg linezolid twice daily.

In the study of Meagher et al. [60], which had a very good to excellent quality (18/19), a dose of 600 mg twice daily orally or i.v. resulted in an AUC_0–24_ of 228 ± 58.4 µg/mL × 24 h from which a C_average_ of 9.5 µg/mL ± 2.4 µg/mL was calculated.

In addition, the study of Beringer et al. [61] had a very good to excellent quality (18/19). The AUC_0–12_ was reported as 62.3 µg/mL × h ± 26.5 µg/ mL × h after a single 600 mg i.v. or oral linezolid. The C_average_ was calculated as 5.2 µg/mL ± 2.2 µg/mL.

The study of Whitehouse et al. [62], also of very good quality (17/19), presented data not only after a single dose but also after repeated dosing of 600 mg i.v. at a steady state [62]. The AUC_0–24_ was 3.33 µg/mL × h (SEM 1.23 µg/mL × h) following 600 mg linezolid i.v. The C_average_ after the first dose was calculated as 6.63 µg/mL. The peak values at a steady state were 2.7 times higher than after the first dose. Hence, we calculated the C_average_ at a steady state as 17.9 µg/mL which was 2.7 times the C_average_ following the first dose.

In two of the population kinetic studies of patients, the resulting values for the AUC and the related C_average_ are explained by the different doses used. In Meagher et al. [60], the dose was 600 mg twice daily with a resulting C_average_ of 9.5 µg/mL and in Beringer et al. [61], the dose was 600 mg with a resulting C_average_ of 5.2 µg/mL. In addition, the C_average_ in the study of Whitehouse et al. [62] is in accordance with these values. Notably, the C_average_ at a steady state is higher, which is relevant for concentration-dependent adverse effects.

The milk concentration of linezolid was measured by HPLC in a 32 year-old healthy volunteer who took a single dose of 600 mg linezolid [63]. A C_max_ of 12.36 µg/mL was attained 2 h post-dosing. The AUC_0–24_ in the milk was 100.79 µg/mL × h and the resulting C_average_ in the milk was 4.2 µg/mL.

Lim et al. [64] reported milk concentrations in a female patient who took linezolid for *S. aureus* bacteremia. The collection of milk commenced 45 h after treatment with 600 mg of oral linezolid, every 12 h, was started. Breast milk concentrations ranged from 3.5 to 12.2 mg/L, the average level being 8.1 µg/mL. Serum linezolid in the newborn infant was below 0.2 µg/mL 4 h after the intake of the 600 mg dose.

Rowe et al. [65] obtained a plasma concentration–time profile in the nursing women after the first dose and the 14th day (at a steady state) of treatment with 500 mg linezolid twice daily [65]. Breast milk samples were determined over a 12 h dosing interval on day 1 and 14. In the milk, the peak concentrations at a steady state were approximately twofold higher than the concentrations after the first dose (9.76 µg/mL vs. 18.73 µg/mL). The plasma C_average_ was 6.16 µg/mL on day 1 and 12.24 µg/mL on day 14.

The results of the three small reports are consistent, indicating that linezolid is excreted in the milk. The results by Sagirli et al. [63] and by Lim et al. [64] match very well with a C_average_ in the milk of 4.2 µg/mL following 600 mg daily and of 8.1 µg/mL following 600 mg twice daily. The roughly twofold higher concentrations in the plasma and milk at a steady state in the study of Rowe et al. [65] are consistent with the findings in Whitehouse et al. [62], showing an increase in concentrations by a factor of 2.7 fold at a steady state compared to the first dose. The AUC (0 to 11 h) in the milk was calculated by trapezoidal rule (by the authors) and the C_average_ in the milk was determined as 6.2 µg/mL after the first dose and 10.8 µg/mL on the 14th day of treatment.

#### 3.5.2. M:P Ratio

The M:P ratio was calculated and resulted in 0.88.

#### 3.5.3. Infant Dose

For the infant exposure, we selected the information from the study by Rowe et al. [65] as the most relevant, taking the C_average_ in the milk on the 14th day of treatment of 10.8 µg/mL and multiplying it by 0.185 L/kg bw. The resulting dose in the infant is 2 mg/kg bw per day which is nearly 15% of the standard therapeutic infant dose according to the WHO guidelines [20] and is fourfold lower than the dose of 600 mg per day in adults (corresponding to 10 mg/kg bw per day).

#### 3.5.4. Adverse Effects

In a review article [66], 47 cases of lactic acidosis, a rare side effect, were identified under treatment with linezolid, from which 12 patients (25.5%) died. No risk factors could be identified [66].

Myelosuppression, a further rare adverse effect of linezolid, was shown to be associated with the metabolite-to-parent concentration ratio of PNU-142300, indicating the role of the toxicity of the metabolite [67].

A threshold of 2 to 8 µg/mL for C_min_ was proposed by Pea et al. [68] and by Matsumoto et al. [69] to minimize thrombopenia due to linezolid.

Extremely rare cases of the serotonin syndrome characterized by high body temperature, confusion, agitation, cardiac arrhythmia, seizures and coma have been described. Linezolid is a reversible, non-selective monoaminoxidase (MAO) inhibitor. Hence, the syndrome may be triggered when patients are concomitantly treated with medications with serotonergic properties such as antidepressants [70]. Because of the occurrence of peripheral and optic neuropathy in rare cases, patients should be advised to report symptoms of visual impairment [71].

No cases of severe adverse effects are reported in breastfed infants. However, the EID is only fourfold below the adult dose and about 15 % of the therapeutic infant dose with high uncertainty. The single value of an isolated plasma concentration <0.2 µg/mL [64] is not plausible given the fact that the M:P ratio is calculated to be 0.88. In addition, due to limited clinical experience with linezolid, concern for adverse effects could not be ruled out (Table 2).

### 3.6. Pretomanid

#### 3.6.1. Kinetics

For pretomanid, several PK studies were identified in the literature search (Table 1); however, no studies were performed on nursing women. Furthermore, no data were published on pretomanid concentrations in human milk or in the breastfed infant to reflect internal exposure. According to the methods section, we selected the available population PK study as the source for the information on the dose-related AUC and calculated the average concentration. The quality of the study by Salinger et al. [72] was rated as excellent (20/20). We selected the AUC after 14 days of continuous treatment as an appropriate parameter. The recommended dosing of pretomanid is 200 mg, four times daily. Although in this study, several doses have been tested, the dose of 800 mg was not among them. In addition, pretomanid exhibits dose-dependent kinetics. Hence, the AUC following 200 mg orally was multiplied by four, simulating the dosing schedule of 200 mg four times per day. The resulting C_average_ resulted in 6.2 µg/mL.

#### 3.6.2. M:P Ratio

Because no concentration in milk is available in the literature for pretomanid, the milk to plasma concentration ratio was calculated using the formulas given in Abduljalil et al. [18] and resulted in 0.89.

#### 3.6.3. Infant Dose:

The external dose for the infant was calculated as 1.02 µg/kg bw per day when a standard dose of 200 mg q.d. is given to the mother.

#### 3.6.4. Adverse Effects

In clinical trials, pretomanid was used together with linezolid and bedaquiline. In this treatment regimen, common adverse effects were anemia, leukopenia, thrombocytopenia, elevated liver enzymes and lactic acidosis. Pretomanid alone can cause dose-dependent QT prolongation which was moderate at a maximum plasma concentration of 6.2 µg/mL with a dose of 400 mg daily. QT prolongation was higher when pretomanid was given in the bedaquiline–pretomanid–linezolid regimen [73], thus increasing the risk for cardiac arrhythmias. However, no arrhythmias have been observed in the clinical trials so far. With respect to the enormous dose difference between the standard dose of 200 mg q.d in adults, roughly 3.3 mg/kg bw, and the EID of 1 µg/kg bw, it is very unlikely to expect adverse effects in the breastfed infant.

For pretomanid, WHO have not made dose recommendations for infants because the data are not yet sufficient.

### 3.7. Pyrazinamide

#### 3.7.1. Kinetics

For pyrazinamide, several PK studies were found in the literature search (Table 1); however, none were performed on nursing women. Data were published on pyrazinamide concentrations in human milk, but not in the breastfed infant to reflect internal exposure. According to the methods section, we selected the two available population PK studies as the source of information on the dose-related AUC and calculated the average concentration. The quality of the study of Maggis-Escurra et al. [74] was rated as excellent (19/20). We selected the AUC as an appropriate parameter which was 380 (range 267–679) µg/mL × h. The resulting C_average_ was 15.8 µg/mL and the dose of pyrazinamide was 28.7 (range 11.7–32.5) mg/kg bw. The second population kinetic study [75] was also rated as very good to excellent (17/18). The AUC was 307 µg/mL × h. The resulting C_average_ resulted in 12.8 µg/mL. The dosing of pyrazinamide was 18 to 27 mg/kg bw daily. The difference in the AUC and, hence, the C_average_ of roughly 20% is explained by the different doses given to the patients.

Holdiness [76] described a case of a 29-year-old woman who received 1 g of pyrazinamide. Milk was collected at 0, 1, 2, 3, 4, 9 and 12 h following oral administration. The quality of this study was rated poor (5/18 points) and no information on the method for the quantification of pyrazinamide was reported. The maximum concentration in the milk was 1.5 µg/mL at 3 h whereas the peak plasma concentration of 42.0 µg/mL was observed at 2 h. No information was given concerning the concentration–time course in the plasma nor in the milk. Using a specifically developed method for measurements in milk, three milk concentrations were determined 2, 4 and 6 h after an oral dose of 1750 mg (somewhat more than the standard dose of 1500 mg) was given to two nursing women [77]. The quality of this study was rated poor (7/19 points) because of a lack of information on the two participants, the clinical setting and the study design. The peak concentrations in the milk are given as 59.2 µg/mL and 38.4 µg/mL.

#### 3.7.2. M:P Ratio

Based on the data from Holdiness [76], the M:P ratio would be calculated as 0.04 whereas using the data from Zuma et al. [77] for the milk concentration and the C_average_ from the study of Magis-Escurra et al. [74], having given the same dose to the study participants, the M:P ratio was 3.75 and 2.43.

#### 3.7.3. Infant Dose

The observed difference between the two publications concerning the milk concentration is not explained by the twofold higher dose in the study of Zuma et al. [77] compared to Holdiness, [76]. Notably, the study by Zuma et al. [77] is not designed originally as a PK study, but rather as an analytical method for determining pyrazinamide levels. Given the advanced methodology used in this recent work, drug concentrations are regarded as more reliable than the older work by Holdiness [76]. Therefore, because of the better analytic quality of the data, we calculated the infant dose using the data provided by Zuma et al. [77] which resulted in an EID of 10.9 and 7.1 mg/kg bw per day, respectively. This external dose is very close to the therapeutic standard dose of 15 mg/kg bw per day in infants <5 kg [20] and the REID would be 0.73 and 0.47 compared to the standard dose. No data were available in the breastfed infant to reflect internal exposure.

#### 3.7.4. Adverse Effects

Rare cases of sideroblastic anemia are reported under therapy with pyrazinamide due to the inhibition of the enzyme catalyzing the first step of hem biosynthesis and pyridoxine should be given routinely as a prophylaxis [78].

Thrombocytopenia is extremely rare whereby the contribution of pyrazinamide remains unclear as the cases occurred with combination therapy in which pyrazinamide was given together with rifampicin and isoniazid [79].

Hepatotoxicity associated with pyrazinamide is suspected to be caused by its metabolites as they were found to be highly correlated to the extent of hepatotoxicity [80].

Given the high exposure in the breastfed infants, adverse effects could not be ruled out (Table 2). The two publications reporting uneventful outcomes in four infants of nursing mothers treated with pyrazinamide are of limited reassurance given the small number of cases [81,82].

### 3.8. Streptomycin

#### 3.8.1. Kinetics

For streptomycin, we found a large number of studies in the literature. However, most of the publications were before 1980, not in the English language and, in addition, most of them were not electronically available (Table 1). One study was performed on nursing women in the Russian language; according to a secondary source (LactMed^®^), the dose was unclear as was the method for the quantification of the concentration in the plasma of the mothers and ambiguous concentration units (not given in mg/L). Data were published on streptomycin concentrations in human milk; however, the primary sources were not accessible as they were published in the Russian and Japanese language. No data were available in the nursed infant to reflect internal exposure.

According to the methods section, we selected the available population PK study as the source for information on the dose-related AUC and calculated the average concentration. The quality of the study of Zhu et al. [83] was rated as good (14/17 points). We selected the AUC as an appropriate parameter which was 225 µg/mL × h. The resulting C_average_ was 9.4 µg/mL. The dosing of streptomycin was 981 mg, corresponding to 16.36 mg/kg bw i.m. A classical PK study was also available [33] which we rated as very good quality (16/18). The AUC was 196.7 ± 25.6 µg/mL × h. The C_average_ resulted in 8.2 ± 1.1 µg/mL for a dose of 1000 mg intramuscularly (corresponding to 14.7 mg/kg bw). The difference in the AUC and, hence, the C_average_ of roughly 15% is explained by the different doses given to the patients.

Holdiness [76] reported on the streptomycin concentration in milk without further information. He stated in a table that the amount ingested by the breastfed infant would be 5 mg. In LactMed^®^, the reference [84] is cited, indicating that 45 women provided milk samples which were analyzed. The analytical method is not mentioned, and the unit of the dose is unclear. Fujimori [85] also provided data on the milk concentrations in eight women, according to LactMed^®^. The concentrations in the breast milk were 0.3 to 0.6 µg/mL, 30 min following injection and 1.1 to 1.3 µg/mL at 6 hours following injection.

#### 3.8.2. M:P Ratio

No reliable data were available to calculate the M:P ratio.

#### 3.8.3. Infant Dose

When calculating the external exposure, by multiplying the concentration at six hours following ingestion with the daily ingested milk volume of 0.185 L/kg bw, EIDs of 0.20 and 0.24 mg/kg bw per day, respectively, are the results.

No data were available in the nursed infant to reflect internal exposure.

#### 3.8.4. Adverse Effects

In patients treated with streptomycin, monitoring of the renal function as well as the auditory and vestibular function is essential [21] as these adverse effects are not uncommon and serious.

However, streptomycin is not nearly absorbed from the gastrointestinal tract [86]. Thus, the internal exposure in breastfed infants is negligible and the probability of adverse effects in breastfed infants is extremely low (Table 2).

### 3.9. Ethambutol

#### 3.9.1. Kinetics

For ethambutol, only two kinetic studies were found in the literature search after 2017, one year before the publication of the study modelling the internal exposure of rifampicin in the nursed infant [16] (Table 1). They were both excluded after screening of the title and abstract. Kinetic data were published before 2017 by Peloquin et al. [87] and McIlleron et al. [88].

#### 3.9.2. M:P Ratio

Before 2017, data on the M:P ratio for ethambutol were published as a personal communication [89], the concentrations being reported as 1.4 mg/L and 4.6 mg/L in the milk and 1.5 mg/L and 4.6 mg/L in the plasma, respectively, from which an M:P ratio of 1 was derived.

After 2017, using a specifically developed method for measurements in the milk, three milk concentrations were determined 2, 4 and 6 h after a dose of 1200 mg, corresponding to 200 mg/kg bw (8.2-fold higher than the standard dose of 24.5 mg/kg bw in [88]), was given to two nursing women [77]. The quality of this study was rated poor (7/20 points). The peak concentrations in the milk are given as 20.7 and 36.4 µg/mL.

#### 3.9.3. Infant Dose

When calculating the external exposure from these milk concentrations by multiplying with the daily ingested milk volume of 0.185 L/kg bw, EIDs of 3.8 and 6.7 mg/kg bw per day, respectively, are the results.

No data were available in the breastfed infant to reflect internal exposure.

In the PK modelling study [16], using the standard dose of 24.5 mg/kg bw, the EID for the breastfed infant resulted as 0.8 mg/kg bw per day. The REID was calculated as 0.05 and 0.03 for the WHO recommended infant dose of 15 to 25 mg/kg bw per day [20]. The difference between this dose and the external infant dose calculated by the data from Zuma et al. [77] is due to the higher dose given to the mothers. Moreover, they used the peak concentrations in their calculations.

In the PBPK study, the concentration–time profile of the breastfed infant was simulated with peak concentrations of roughly 0.1 µg/mL, 3.7-fold lower than in the nursing mothers.

#### 3.9.4. Adverse Effects

Severe adverse effects of ethambutol are rare. The most severe adverse effect is dose-dependent optic neuropathy with the symptom of red/green color blindness [90]. The prevalence of optic neuropathy was lower when doses below 30 mg/kg bw per day were used [90]. The dose-dependency of optic neuropathy was confirmed in recent studies of Korean and Taiwanese patients [91,92].

Even though cases of optic neuropathy were observed in Korean patients with doses as low as 12.3 mg/kg bw per day, it seems safe to say that optic neuropathy is not expected to occur at a dose as low as 0.8 mg/kg bw per day which we estimated for the breastfed infant (Table 2). In addition, the relative infant dose was low (0.05 and 0.03), confirming no health concerns.

### 3.10. Rifampicin

#### 3.10.1. Kinetics

For rifampicin, only a few PK studies were found in the literature search after 2017, one year before the publication of the study modelling the internal exposure of rifampicin in the breastfed infant [16] (Table 1).

#### 3.10.2. M:P Ratio

Before 2017, published data on the milk/plasma ratio for rifampicin are scarce [38,93]. The milk concentration was reported as 3.4–4.9 mg/L and the plasma concentration as 21.3 mg/L [93], and the other was 10–30 mg/L in the milk and 50 mg/L in the plasma [38]. The ratio was 0.16–0.23 in the first data pair, and 0.2–0.6 in the second. As the reported data are extracted from review articles, it is unclear at which time after drug intake the samples were obtained.

#### 3.10.3. Infant Dose

After 2017, using a specifically developed method for measurements in the milk, three milk concentrations were determined 2, 4 and 6 h after a dose of 900 mg (twice the standard dose) was given to two nursing women [35,77]. The quality of this study was rated poor (7/21 points). Peak concentrations in the milk are given as 14.0 µg/mL and 17.6 µg/mL. When calculating the external exposure by multiplying with the daily ingested milk volume of 0.185 L/kg bw, doses of 2.6 and 3.3 mg/kg bw per day, respectively, are the result. The REID is 0.17 and 0.22, respectively, compared to the WHO recommended infant dose of 15 mg/kg bw per day [20]. No data were available in the breastfed infant to reflect internal exposure after twice the standard dose.

In the PBPK study [16], using the standard dose of 450 mg, the external dose resulted as 0.4 mg/kg bw. The difference between this dose and the external infant dose calculated by Zuma et al. [77] is due to the higher dose given to the mothers and also because they used the peak concentrations for their calculation.

In the PK modelling study, the internal exposure of the breastfed infant was simulated with peak concentrations of roughly 2 µg/mL, tenfold lower than in the nursing mothers.

#### 3.10.4. Adverse Effects

Severe adverse effects of rifampicin are rare. Hematological adverse effects may occur as agranulocytosis, hemolytic anemia and thrombocytopenia, the latter in most cases with high-dose therapy. Increased liver enzymes may occur due to the induction of metabolizing liver enzymes (Cytochrome p 450, especially CYP 3A4 and glucuronosyltransferases) and p-glycoprotein activities [94].

For rifampicin, the lowest recommended dose in infants is 10 mg/kg bw per day which is 25 times higher than the dose of 0.4 mg/kg bw per day we estimated to which the breastfed infant is exposed. A dose of 600 mg rifampicin per day in an adult (8.2 mg/kg bw per day) is typically used to investigate the induction of cytochrome enzymes (CYPs), mainly CYP 3A4 [95,96]. This dose is also more than 20 times higher than the dose transferred via breast milk to the infant. The relative infant dose ranged from 0.17 and 0.22. Consequently, it can be concluded that exposure to rifampicin through nursing is safe for the breastfed infant (Table 2).

### 3.11. Isoniazid

#### 3.11.1. Kinetics

For isoniazid, only a few PK studies were found in the literature search after 2017, one year before the publication of the study modelling the internal exposure of rifampicin in the nursed infant [15] (Table 1).

#### 3.11.2. M:P Ratio

Before 2017, data on the milk concentrations and M:P ratio for isoniazid were published [97,98,99,100]. The quality of the older studies, according to contemporary standards, is poor, mainly because details of the study and the analytical method used are not reported. In the study of Singh et al. [100], using a method that is also valid for measurements in the milk, the plasma and milk concentrations were determined up to 24 h following the ingestion of 300 mg isoniazid in the morning in seven nursing women on a continuous therapy with isoniazid 300 mg, rifampicin 450 mg and ethambutol 800 mg [100]. The quality of this study was rated as fair (12/16 points). After 2017, no specific studies measuring the concentrations in the milk were published.

The mean M:P ratio of isoniazid calculated by dividing the AUC_milk_ by the AUC_plasma_ was 0.89 (95% CI 0.7, 1.1) [100].

#### 3.11.3. Infant Dose

When calculating the external exposure by multiplying with the daily ingested milk volume of 0.185 L/kg bw, an EID of 0.11 mg/kg bw per day is the result. The REID was calculated as 0.007 for the WHO recommended dose of 15 mg/kg bw per day [20]. No data were available for the nursed infant.

A PBPK study, validated by the experimental data of Lass and Bünger [97], Ricci and Copaitich [98] and Singh et al. [100], allowed to detangle the influence of the slow vs. fast acetylator type of isoniazid metabolism in nursing mothers combined with the slow vs. fast acetylator type of isoniazid metabolism in the breastfed infant on the external as well as the internal exposure in the breastfed infant [15]. In the PK modelling study, using the standard dose of 300 mg, the plasma and the milk C_average_ were doubled in the mothers who were slow metabolizers (fast vs. slow metabolizer 0.82 µg/mL vs. 1.82 µg/mL and 0.72 µg/mL vs. 1.59 µg/mL for the plasma and milk, respectively) and the resulting EID was 0.13 mg/kg bw per day if the mother was a fast metabolizer of isoniazid and 0.3 mg/kg bw per day if the mother was a slow metabolizer of isoniazid. The internal exposure of the nursed infant was simulated with average concentrations of 0.01 µg/mL in the combination of a fast-metabolizing mother and a fast-metabolizing infant, 0.04 µg/mL in the combination of a fast-metabolizing mother and a slow-metabolizing infant, 0.03 µg/mL in the combination of a slow-metabolizing mother and a fast-metabolizing infant and 0.1 µg/mL in the combination of a slow-metabolizing mother and a slow-metabolizing infant. Thus, the influence of the acetylator type on the internal isoniazid exposure in the infant shows a tenfold higher peak concentration in the plasma of the breastfed infant for the situation when both the mother and infant are slow metabolizers compared to the situation when both the mother and infant, are fast metabolizers.

#### 3.11.4. Adverse Effects

Severe adverse effects of isoniazid are rare. However, peripheral neuropathy is a common finding in patients. Single cases of agranulocytosis and thrombocytopenia and drug-induced lupus erythematosus have been reported [101].

As the C_average_ in the infants of the slow metabolizer type, having a mother with a slow metabolizer type, and, therefore, the most exposed, it could be assumed that the risk of adverse effects due to isoniazid in breastfed infants is low except for lupus erythematosus as this is a concentration-independent, immunological effect (Table 2). However, no cases of lupus in infants have been reported so far in the literature.

## 4. Discussion

Lactating mothers represent an important subpopulation among the patient population with tuberculosis. They contribute to a significant proportion of the global TB burden and require optimized TB treatment. Because of its multiple benefits for both the mothers and infants, breastfeeding has been recommended by official authorities worldwide, and according to epidemiological studies, is increasingly practiced [102]. Breastfeeding is particularly important in developing countries where resources are limited as it provides essential nutrients and is cost-effective. Promoting and supporting breastfeeding in low-income settings where TB is endemic is crucial for improving the health and well-being of both mothers and infants [9]. A robust assessment of infant exposure to anti-TB drugs through breast milk and the potential of eliciting adverse effects in the breastfed infant are of utmost importance since mothers should be treated adequately with the full therapeutic doses of drugs. The latter will pass into the breast milk and the breastfed infant and might be associated with adverse reactions. Although the knowledge about medication use during lactation has largely been expanded in recent years, the potential of adverse effects caused by several drugs in breastfed infants remains to be elucidated. The situation is even more critical with many of the new drugs, where adverse effects have not been sufficiently studied yet which applies to some anti-TB agents such as bedaquiline and pretomanid. The overall safety of a drug in a breastfed infant depends not only on the extent of the external exposure to a drug determined by the drug properties (molecular weight, lipid solubility, pH and protein binding), the extent of feeding and maternal disposition but also the capability of the infant to metabolize and eliminate a drug which is not yet fully developed at birth and is dependent on the gestational age. Even if the external exposure dose is only a fraction of the maternal dose, estimating the systemic exposure of the infant is more consequential to evaluate the risk of developing adverse effects due to the risk of accumulation. It is more appropriate to compare the external dose of the infant with that of a treated infant, resulting in a relative external infant dose. Another important aspect to consider is the variability in the pharmacokinetics and pharmacodynamics of anti-TB agents as a result of polymorphisms in genes encoding for drug metabolizing enzymes and membrane transporter proteins relevant for these drugs [103]. This is especially important for the safety of anti-TB drugs in the mother and the breastfed infant since they are used for lengthy treatments. For example, the ototoxicity of aminoglycosides has been suggested to relate to mutations in mitochondrial DNA, in particular in the 12S rRNA genes such as the 1555A> G and 1494C > T mutations which predisposes subjects carrying them to an impairment of protein synthesis that might lead eventually to hearing loss [104]. Fluoroquinolones are substrates of multiple ABC transporters [105] and for levofloxacin, reduced activity of P-glycoprotein and BCRP might increase the risk for neurological adverse effects due to the enhanced penetration of levofloxacin through the blood–brain barrier [106]. Similarly, for moxifloxacin, polymorphisms of the SNPs rs8175347 and rs3755319 in UGT1A as well as in the ABCB1 SNP rs2032582 were associated with altered moxifloxacin clearance and bioavailability [107]. Furthermore, variants in the SLCO1B1 gene were associated with higher moxifloxacin exposure in TB patients [108]. As for Linezolid, the polymorphism (3435C > T) in SNP rs1045642 in the ABCB1 gene, coding for P-glycoprotein, was significantly associated with linezolid clearance [109].

In this review, we focused on three areas: (1) assessing the available literature on the kinetics of anti-TB drugs using a published scoring system; (2) evaluating the current literature available on the excretion of anti-TB drugs in breast milk and the subsequent drug exposure of the infant via breast milk; and (3) the probability of occurrence of adverse effects in the breastfed infant secondary to the estimated exposure. Based on the WHO list of anti-TB drugs [8], we examined the first-line anti-TB drugs as well as new agents including bedaquiline and pretomanid. We performed a literature search for the published studies on the PK of the selected drugs, extracting relevant information such as the maternal average concentrations, AUC and concentrations measured in the milk following exposure to therapeutic doses of the drugs.

Surprisingly, the studies describing PK in treated nursing mothers were extremely limited and were only found for 3/11 drugs (linezolid, bedaquiline and clofazimine). On the other hand, PK data in the patients/general population were satisfactory based on the scoring system that we used. In contrast, data describing the drug concentration in breast milk were satisfactory for only 3/11 drugs: linezolid, clofazimine and pretomanid, as outlined by Table 2. In most of the cases, the information was not satisfactory because of its limited quality and the small number of subjects included in the studies. Hence, the assessment of the infant external exposure, for which the milk concentration is needed, is surrounded by some, in some cases even high, uncertainty. Extremely limited data on infant plasma concentrations were available for linezolid and bedaquiline. For bedaquiline, a single sample was measured in an infant while the data for linezolid were unreliable.

Given the fact that experimental concentration–time profiles in breastfed infants are difficult to obtain and not available in the literature, PBPK models describing the exposure of the infant by breastfeeding could help to improve the understanding of drug pharmacokinetics in lactating women and estimate the amount of drug ingested by the infant during breastfeeding, which is a helpful approach when the data are not fully satisfactory. Moreover, the plasma concentration–time profile could be simulated in the infant, enabling the modelling of systemic exposure considering the immaturity of the eliminating organs. This was successfully undertaken for some drugs such as ethambutol, rifampicin and isoniazid which contributed to the synthesis of recommendations regarding the safety of their use during lactation, as shown in this review.

The assessment of potential adverse effects was performed by comparing the external infant exposure dose to the therapeutic infant dose. In cases when we had concentration-dependent adverse effects, the assessment was based on the modelled concentrations in the infants, if available.

For clofazimine, ethambutol, levofloxacin, isoniazid and rifampicin, our assessment came to the conclusion that there is no concern for the breastfed infant when mothers are treated with the full therapeutic dose, based on the low relative infant exposure dose (clofazimine: 0.05–0.12; ethambutol: 0.03–0.05; levofloxacin: 0.05 and 0.06; isoniazid: 0.007; rifampicin: 0.17–0.22). This is in line with recommendations by several authors. Ethambutol and isoniazid have been classified by the American Academy of Pediatrics to be compatible with breastfeeding [110]. However, for some of the drugs, monitoring of the infants for clinical signs and symptoms is recommended. For clofazimine, it is noted that discoloration of the skin may occur (LactMed^®^). Because of a possible influence on the gastrointestinal flora, infants whose nursing mothers receive levofloxacin should be monitored for diarrhea, thrush or diaper rash (LactMed^®^). In addition, rifampicin is classified as compatible with breastfeeding; however, monitoring the infant for jaundice is recommended [110].

We did not find a publication dealing with the use of pretomanid in breastfeeding women with respect to the safety of the infant. The WHO has not established a dose recommendation in infants yet; therefore, we had to base our assessment on the standard adult dose. Comparing the external infant dose with the adult dose resulted in a difference of more than 3000-fold. Hence, it is safe to say that there is no health concern for the breastfed infant.

For streptomycin, a health concern for the infant could be ruled out owing to the extremely low oral bioavailability [86].

In contrast to the positive recommendations for other drugs, e.g., bedaquiline, some concern exists for potential adverse effects in the infant based on the level of data available. In the case of bedaquiline, this is due to the high relative infant dose (0.65) and the lack of safety data in infants (children < 6 years). In the SPC of the European Medicines Agency, the following sentence states the regulatory view on the use of this drug in breastfeeding women, “Because of the potential for adverse reactions in breastfed infants, a decision must be made whether to discontinue breast-feeding or to discontinue/abstain from bedaquiline therapy taking into account the benefit of breast-feeding for the infant and the benefit of therapy for the mother” (Summary of Product Characteristics, European Medicines Agency, last updated 23/11/2021).

For other drugs, data were extremely limited to rule out concern for adverse effects such as cycloserine/terizidone and pyrazinamide. In the case of linezolid, the external infant dose is only fourfold below the adult dose and about 15 % of the therapeutic infant dose added to the limited clinical experience.

For pyrazinamide, infants should be monitored for the rare side effects observed in adult patients such as clinical signs for thrombocytopenia and of liver toxicity (loss of appetite, nausea and vomiting, jaundice) LactMed^®^. For other anti-TB drugs, no authoritative statement was found.

Measuring the drug concentration in human milk represents the most accurate basis for estimating the infant external exposure, factoring in the amount of milk consumed by the infant. Although measuring the drug concentration in milk samples is considered a non-invasive technique, the implementation of a clinical study is frequently hampered by many obstacles, such as nursing women enrollment, transportation issues, milk sample collections and analysis, especially in a low-income setting which obscures the quality of the needed data. For a more robust assessment, concentration data in relevant biological fluids such as plasma or urine would allow for estimating the internal exposure in infants. However, this poses ethical and methodological challenges.

## 5. Conclusions

In most of the cases, data were not satisfactory for a proper estimation of the external infant exposure regarding the measured drug concentration in breast milk. The vast majority of the published data suffer from limitations in sample collection, timing and number of participants as well as in the study design. Though extremely helpful in evaluating the potential of adverse effects in the infant, plasma concentration data are extremely scarce in the literature and the values are sometimes uncertain. Furthermore, very little published data exist documenting the clinical outcome in exposed infants. Based on our analysis, concern for potential adverse effects in breastfed infants could not be ruled out for bedaquiline, cycloserine/terizidone, linezolid and pyrazinamide.

## 6. Future Directions

Adequate studies of treated nursing mothers should be performed which entail methodologically sound measurements of breast milk concentrations in relation to drug-dosing regimens following standardized protocols for specimen collection and analysis. Measuring concentration–time profiles in the milk of patients is not an invasive technique and could be technically well performed. The concentration–time profiles of the mothers as well as internal exposure data in breastfed infants are extremely helpful for assessing the potential of adverse effects. Less invasive techniques rather than blood sampling could be considered to be applied in both newborn infants and lactating mothers. These could include the measurement of drug concentrations in saliva which was previously described for caffeine [111] and recently for levofloxacin [48]. The measurement of drug amounts excreted in urine as described for bisphenol A in infants in an intensive care unit [112] could serve as a rough approximation of internal exposure. Financial support should be provided to perform such studies with high quality in the target population.

## Figures and Tables

**Figure 1 pharmaceutics-15-01228-f001:**
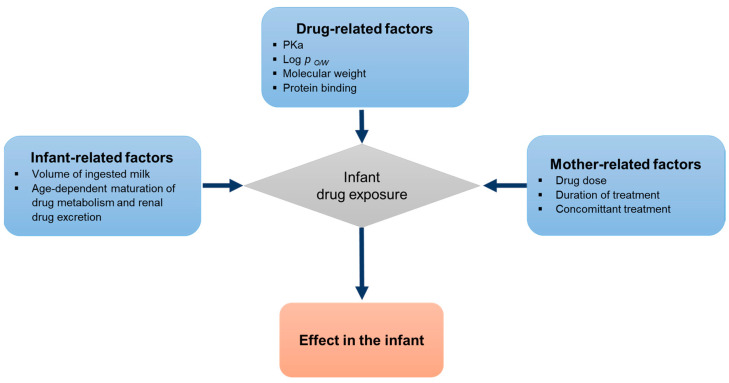
Factors influencing infant drug exposure by nursing.

**Figure 2 pharmaceutics-15-01228-f002:**
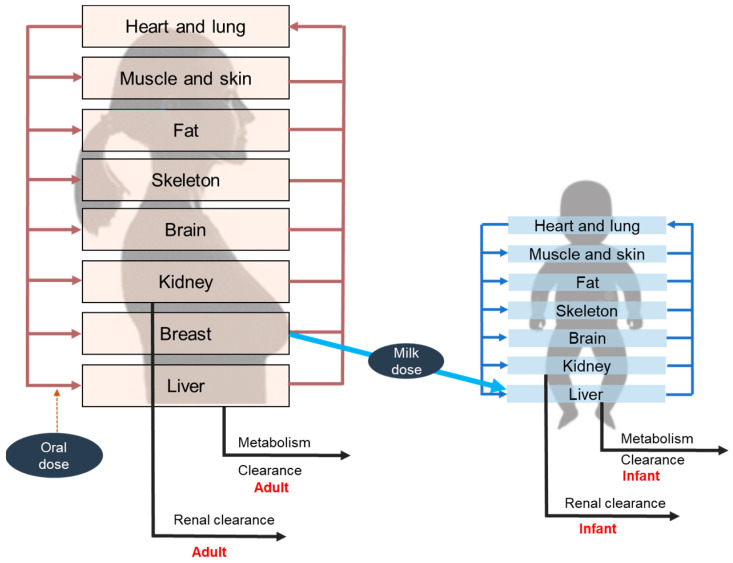
Conceptual representation of physiologically based pharmacokinetic (PBPK) model for the lactating mother and the breastfed infant.

**Figure 3 pharmaceutics-15-01228-f003:**
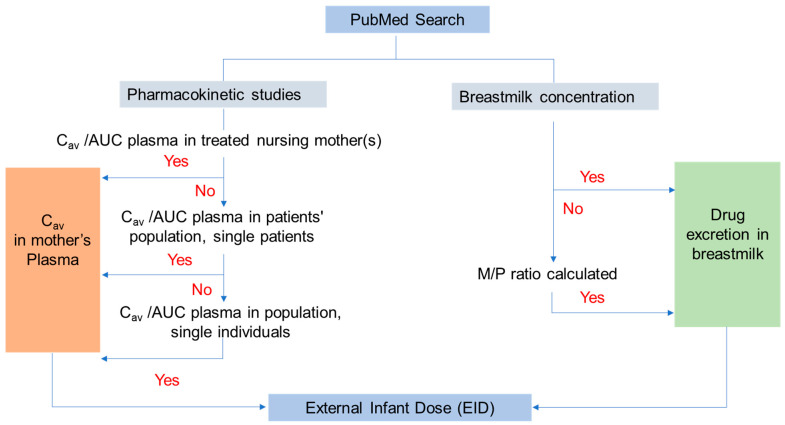
A schematic flowchart showing the search strategy and milk data collection from PubMed. C_av_, average plasma concentration; AUC, area under the concentration–time curve.

**Table 1 pharmaceutics-15-01228-t001:** Overview on (A) number of articles found in the search, (B) number of articles remaining after title and abstract screening and (C) number of articles remaining after full text screening.

Anti-TB Drug	Retrieved Articles								
	Data on Plasma Concentration			Data on Milk Concentration			Data on Plasma Concentration in Breastfed Infant		
	A	B	C	A	B	C	A	B	C
PK studies, including population kinetics									
Bedaquiline	7	3	1	1	1	1	1	1	1
Clofazimine	1	0	0	1	0	0	0	0	0
Cycloserine /Terizidone	17	9	5	0 indirect information, no access to the original publication	0 indirect information, no access to the original publication	0 indirect information, no access to the original publication	0	0	0
Levofloxacin	31	12	6	2	1	1	6	0	0
Linezolid	53	7	3	6	3	3	11	3	3
Pretomanid	14	4	1	0	0	0	0	0	0
Pyrazinamide	110	7	5	3	2	0	19	0	0
Streptomycin	332 ^1^	5	2	2 (Japanese language, Russian language)			8	0	0
PK studies, modelling approaches									
Ethambutol	2 ^2^	0	0	1	1	1	6	1 ^3^	n.a.
INH	22 ^2^	2	0	No new data after 2017			9	1 ^4^	n.a.
Rifampicin	141 ^2^	3	0	7	1	1	14	1 ^3^	n.a.

^1^ most of the studies are not in English, before 1975 not electronically available; ^2^ since 2017; ^3^ simulated by Partosch et al. [16]; ^4^ simulated by Garessus et al. [15]; n.a., not applicable.

**Table 2 pharmaceutics-15-01228-t002:** Overview on the results of the assessment.

Anti-TB Agent	PK Studies		Breast Milk Concentration Data	Plasma Concentration Data in Breastfed Infants	Comment on Adverse Effects
PK studies, including population kinetics					
	Nursing women	patients/general population			
Bedaquiline	Yes	Limited	Limited	Yes ^1^	Some concern ^3^
Clofazimine	Yes	Not satisfactory	Satisfactory	No data	No concern
Cycloserine /Terizidone	No	Satisfactory	Not satisfactory	No data	Concerns could not be ruled out
Levofloxacin	No	Satisfactory	Not satisfactory	No data	No concern
Linezolid	Yes	Satisfactory	Satisfactory	Yes ^2^	Concerns could not be ruled out
Pretomanid	No	Satisfactory	Satisfactory	No data	No concern based on the 3000-fold difference between the standard adult and external infant dose
Pyrazinamide	No	Satisfactory	Not satisfactory	No data	Concerns could not be ruled out
Streptomycin	Limited (Russian language, dose unclear)	Satisfactory	Not satisfactory (secondary sources, primary sources in Russian and Japanese)	No data	No concern as streptomycin is orally not bioavailable
PK studies, modelling approaches					
Ethambutol ^4^	No	Satisfactory	Not satisfactory	Simulated	No concern
Rifampicin ^4^	No	Satisfactory	Not satisfactory	Simulated	No concern
Isoniazid ^5^	No	Satisfactory	Not satisfactory	Simulated	No concern

PK, pharmacokinetic. ^1^ only one sample; ^2^ doubtful result; ^3^ based on high relative infant dose (0.65), lack of safety data in infants, prolongation of QTc, a potential life-threatening side effect in adults; ^4^ Partosch et al. [16]; ^5^ Garessus et al. [15].

## Data Availability

Data sharing not applicable. No new data were created or analyzed in this study.

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
