# Peer review of "Infant Exposure to Antituberculosis Drugs via Breast Milk and Assessment of Potential Adverse Effects in Breastfed Infants: Critical Review of Data"

_pharmaceutics, 2023, doi:10.3390/pharmaceutics15041228_

Round 1

Reviewer 1 Report

The authors evaluated the quality of existing data on antituberculosis drugs concentrations in plasma and breast milk as a methodologically sound basis for the potential risk of breastfeeding while undergoing tuberculosis therapy.

I would recommend the following minor corrections:

Abstract:

Line 27 – Replace “rule out” with ruled out.

Line 18 – “methodological sound basis for the potential risk of breastfeeding under therapy” – please check, methodically seems more appropriate

Introduction:

Line 47 – “outcomes for both pregnant women and infants. A global disease burden amounting to 216,500 cases was 47 estimated in 2011 in pregnant and postpartum women” – Please add more recent statistics here. 2011 is a long while ago.

Body:

Line 57: “it is said to be safe for the infant” – Delete extra space between it and is.

Line 132: “If references were found they were added to the reference list”. – This sentence should be rewritten. The meaning is not clear.

Line 552: the nature of the combination therapy should be specified.

Line 555: I would suggest deleting the second pyrazinamide and replacing with its.

Author Response

Abstract:

Line 27 – Replace “rule out” with ruled out.

Thank you for the correction. Changes have been made accordingly.

Line 18 – “methodological sound basis for the potential risk of breastfeeding under therapy” – please check, methodically seems more appropriate.

Thank you for the correction. Changes have been made accordingly.

Introduction:

Line 47 – “outcomes for both pregnant women and infants. A global disease burden amounting to 216,500 cases was 47 estimated in 2011 in pregnant and postpartum women” – Please add more recent statistics here. 2011 is a long while ago.

We agree with the Reviewer that the statistic pertains to a long while ago. However, in light of the recently provided data from the WHO (World Health Organization, 2023. WHO recommendations on antenatal care for a positive pregnancy experience: screening, diagnosis and treatment of tuberculosis disease in pregnant women.), which reportedly suggest the deficiency in globally recording the prevalence of TB in pregnant and postpartum women and refers to the study by Sugerman et al., (Sugarman J et al. Lancet Glob Health. 2014;2(12):e710–6) as the source for an estimated prevalence especially in pregnant women. In their investigation, the authors used publicly accessible country level estimates of demographic and epidemiological parameters from 217 countries to derive estimates of the number of pregnant women with active tuberculosis. Indeed, the neglected global burden of tuberculosis in pregnant women and its exact magnitude have been previously discussed (Zumla, A. et al. The Lancet Global Health, 2014; 2, pp.e675-e676.). Given the actual burden of pregnant women with active TB is not currently known, an estimation was proposed in the literature (Mathad JS et al. Clin Infect Dis 2012;55:1532–49.) for the prevalence depending on the WHO region which is supported by individual reports on the prevalence from developed (Nordholm AC et al. Eurosurveillance, 2022;27(12), p.2100949.) and low- and middle-income countries countries (Pasipamire M et al. African journal of laboratory medicine, 2020;9(1), pp.1-9.). A recent study in Mozambique (Nguenha D, et al. Int J Tuberc Lung Dis. 2022;26(7):641-649) reported a TB prevalence of (0.5%) and (0.3%) among pregnant and post-partum women, respectively.  Another study in Benin reported a prevalence of 0.049% (Adjobimey M, et al. PLoS ONE, 2022;17(2): e0264206).

To clarify this point, we added a new paragraph on the burden and the estimated prevalence of TB in pregnant women lines 40-46 as follows:

“The actual global burden of pregnant women with active TB is not currently known, hence, an estimation of the prevalence is given in the literature depending on the WHO region which ranges from 0.06% to 0.25% in low-burden countries to 0.07% and 0.5% in high-burden countries among HIV-negative women (Bates et al., 2015). This is also supported by individual reports on the prevalence emerging from developed countries (Nordholm et al., 2022) and developing/low-income countries (Pasipamire et al., 2020). A recent study in Mozambique (Nguenha et al. 2022) reported a TB prevalence of (0.5%) and (0.3%) among pregnant and post-partum women, respectively.  Another study in Benin reported a prevalence of 0.049% (Adjobimey et al., 2022)."

Body:

Line 57: “it is said to be safe for the infant” – Delete extra space between it and is.

Thank you for the correction. Changes have been made accordingly.

Line 132: “If references were found they were added to the reference list”. – This sentence should be rewritten. The meaning is not clear.

We agree with the Reviewer, hence we modified this part to clarify the meaning so it reads now lines 125-127.

“Furthermore, we checked the LactMed® database to identify additional, potentially relevant, records which were not retrieved by the literature search. New records were then added to supplement the retrieved articles form PubMed searching.”

Line 552: the nature of the combination therapy should be specified.

Thank you for the suggestion. We added the following to line 556:

“…in which pyrazinamide was given together with rifampicin and isoniazid”

Line 555: I would suggest deleting the second pyrazinamide and replacing with its.

Thank you for the correction. Changes have been made accordingly.

Thank you for all the effort!

Reviewer 2 Report

Review of manuscript entitled: „Infant Exposure to antituberculosis drugs via breast milk and assessment of potential adverse effects in breastfed infants: critical review of data”. The paper presents a review of the literature on antituberculosis drugs found in plasma and milk for the potential risk of breastfeeding under therapy. The Authors performed a systematical search in PubMed database, and use some keywords. In my opinion, the method and results section is detailed. The work is valuable and contains important information. The identified gaps in knowledge indicate a field for further research.

Comments and suggestions:

1. Improve the aesthetics of the tables, please

2. Improve the aesthetics of the captions under the drawings, please

3. Table 2 – is it possible to get the number of „patients/general population”? What does it it mean „Not satisfactory, satisfactory, limited”?

1.      Table 2 should be placed under all drug descriptions, not before „PK studies, modelling approaches”.

Author Response

  1. Improve the aesthetics of the tables, please

Thank you for your suggestion. The lay out of all Tables has been adjusted according to the template provided by the journal.

  1. Improve the aesthetics of the captions under the drawings, please

Thank you for your suggestion. The captions have been improved.

  1. Table 2 – is it possible to get the number of „patients/general population”? What does it it mean „Not satisfactory, satisfactory, limited”?

The number of patients/general population on which our assessment is based is outlined in the text.

Based on the type of data, the quality was assessed pragmatically using the terms “satisfactory” or “not satisfactory”. For PK data, we used the scoring system according to (Gafar et al., 2022) as indicated in the Methods section for rating the PK studies in TB to describe the state of state of knowledge regarding the PK of the drugs as either satisfactory and eligible for using the data further in the calculations of the maternal exposure and hence the infant’s exposure or not due to limitations in the robustness of data. As for the drug concentration in breast milk data, several factors were evaluated including the plausibility of the provided milk data in relation to the dose given to the mother, the number of subjects, the number of samples (concentrations data points) and the timing of sample collection in the studies to judge the robustness and the sufficiency of data to calculate the infant’s exposure.

In order to clarify the terms we used, we added a new sentence to the Methods below:

Lines 146-147:

“The quality of the available data from PK studies was thus described according to the calculated score as either “satisfactory” or “not satisfactory”.

As for the breast milk data, lines 158-161 were added:

“The quality of the available data was evaluated using several factors such as the plausibility of the provided milk data in relation to the dose given to the mother, the number of subjects, the number of samples (concentration data points) and the timing of sample collection in the studies, hence, data were qualified as “satisfactory” or “not satisfactory”.

           Table 2 should be placed under all drug descriptions, not before „PK studies, modelling approaches”.

      Thank you for your suggestion, we relocated Table 2 to after the results of all drugs and before the discussion.

Thank you for all the effort!

Reviewer 3 Report

In this manuscript, the authors summarized recently published data on antituberculosis (anti-TB) drug concentrations for the potential risk of breastfeeding under therapy. Minor revision is recommended, and the comments are as below:

1.     Line 129 “Time constraints (December 2017 until December 2022) were used”, please explain why the authors chose these time constraints.

2.     For the discussion of the “Kinetics” of drugs, for example, lines 310-320, the data is better demonstrated in a table.

3.     The format of references needs to be carefully checked, such as the pagination of ref. 31, 56, and 72 are missing.

Author Response

  1. Line 129 “Time constraints (December 2017 until December 2022) were used”, please explain why the authors chose these time constraints.

We used the time constraints above because a systematic search has been already performed for the relevant studies for the anti-tuberculosis agents rifampicin, ethambutol and isoniazid in two previous publications of one of the authors of the current study which are cited as (Partosch et al., 2018; Garessus, et al., 2019). These records were already considered and accounted for by the two cited previous publications which we used in our analysis. New data published after the date of the performed search in these two publications i.e., 2017, were subsequently searched in PubMed.

  1. For the discussion of the “Kinetics” of drugs, for example, lines 310-320, the data is better demonstrated in a table.

Thank you for the suggestion. We were rather of the opinion that it serves the reader more to present the arguments in full text which allows to explain the underlying assessment in detail.

  1. The format of references needs to be carefully checked, such as the pagination of ref. 31, 56, and 72 are missing.

      Thank you for your comment and keen observation. The references were corrected.

Thank you for all the effort!

Reviewer 4 Report

This manuscript addresses the very important topic of infant exposure to tuberculosis drugs and is very timely and relevant due to the increasing number of Mycobacterium infections and the inability to stop the tuberculosis epidemic by abandoning mass vaccination. I have no concerns with the content of the article, which is very well-prepared with one exception. The authors did not address the topic of pharmacogenetics and a very important aspect regarding the potential side effects of antibiotics (see for example PMC7504675).

Author Response

This manuscript addresses the very important topic of infant exposure to tuberculosis drugs and is very timely and relevant due to the increasing number of Mycobacterium infections and the inability to stop the tuberculosis epidemic by abandoning mass vaccination. I have no concerns with the content of the article, which is very well-prepared with one exception. The authors did not address the topic of pharmacogenetics and a very important aspect regarding the potential side effects of antibiotics (see for example PMC7504675).

We agree on the valid point raised by the Reviewer that this topic should be addressed in context of the safety of anti-TB agents in both the mother and the nursed infant. Therefore, we added the following paragraph to the Discussion section lines 751-765.

“Another important aspect to consider is the variability in the pharmacokinetics and pharmacodynamics of anti-TB agents as a result of polymorphisms in genes encoding for drug metabolizing enzymes and membrane transporter proteins relevant for these drugs (Stocco et al., 2020). This is especially important for safety of anti-TB drugs in the mother and the breastfed infant since they are used for lengthy treatments. For example, the ototoxicity of aminoglycosides have been suggested to relate to mutations in mitochondrial DNA in particular in the 12S rRNA genes such as the 1555A> G and 1494C > T mutations which predisposes subjects carrying them to an impairment of protein synthesis that might lead eventually to hearing loss (Guan et al., 2001).  Fluoroquinolones are substrates of multiple ABC transporters (Alvarez et al., 2008) and for levofloxacin, reduced activity of P-glycoprotein and BCRP might increase the risk for neurological adverse effects due to enhanced penetration of levofloxacin through the blood brain barrier (Gervasoni, et al., 2013). Similarly, for moxifloxacin, polymorphisms of the SNPs rs8175347 and rs3755319 in UGT1A as well as in the ABCB1 SNP rs2032582 were associated with altered moxifloxacin clearance and bioavailability (Naidoo et al., 2018). Furthermore, variants in the SLCO1B1 gene were associated higher moxifloxacin exposure in TB patients (Weiner et al., 2018). As for Linezolid, the polymorphism (3435C > T ) in SNP rs1045642 in the ABCB1 gene, coding for P-glycoprotein, was significantly associated with linezolid clearance (Allegra et al.,2018)”.

Reviewer 5 Report

The manuscript entitled "Infant Exposure to Antituberculosis Drugs via Breast Milk and Assessment of Potential Adverse Effects in Breastfed Infants: Critical Review of Data" Title, abstract and overall rationale of work is well written. However, there are still some major concerns, which needs to be addressed before publication

1) In the keywords: First letter should be capital letter.

2) Introduction section is written well and explain details about the TB, drugs, pregnant women and infant. However, this section written very details and I suggest to author to reduce the introduction section and write concise way.

3) In the method section: Author need to give one flowchart should be added to the article to show the research methodology. For example how many anti-TB drugs related papers author found in different website, how many selected etc etc.

4) Results section: This section written details and lengthy however I recommend to author they need to provide table in each drug activities like rating, calculation and others (for example Bedaquiline and other drugs).

5) In the table 2 author wrote about the Bedaquiline drug adverse effect (Some concern) I suggest author to write details which kind of concern they found.

6) Author need expend discussion and much more explanations and interpretations must be added for the results, which are not enough at all. It is suggested to compare the results of the present research with some similar studies which is done before.

7) There are some of punctuation and typographical errors throughout in the manuscript. kindly correct it

Author Response

  • In the keywords: First letter should be capital letter.

       Thank you for the correction. Changes have been made accordingly.

  • Introduction section is written well and explain details about the TB, drugs, pregnant women and infant. However, this section written very details and I suggest to author to reduce the introduction section and write concise way.

We highly appreciate the suggestion of the Reviewer and indeed shortened the Introduction and reworded some parts in the revised manuscript.

3) In the method section: Author need to give one flowchart should be added to the article to show the research methodology. For example how many anti-TB drugs related papers author found in different website, how many selected etc etc.

Although this information is given already in Table 1, we produced for the kinetics, milk concentration and infant’s plasma concentration data combined a new Flowchart since the literature search was performed for each of these three search strings separately as indicated in the Methods section. The flowchart is now given as a Supplementary Figure 1.

4) Results section: This section written details and lengthy however I recommend to author they need to provide table in each drug activities like rating, calculation and others (for example Bedaquiline and other drugs).

Thank you for the suggestion. We are rather of the opinion that it serves the reader more to have the arguments presented in full text to better explain our assessment, therefore, we respectfully, prefer the results sections as it is.

5) In the table 2 author wrote about the Bedaquiline drug adverse effect (Some concern) I suggest author to write details which kind of concern they found.

Indeed, this point might not be clear to the reader, therefore, we explained in the footnote of Table 2 our reasons for concern which are as follows:

"High relative infant dose (0.65), lack of safety data in infants, prolongation of QTc and the presence of a potential life-threatening side effect in adults."

6) Author need expend discussion and much more explanations and interpretations must be added for the results, which are not enough at all. It is suggested to compare the results of the present research with some similar studies which is done before.

We added to the discussion comparison to recommendations and work done which shows that our assessment is broadly in line with what other authors have claimed. However, for some of the tuberculosis drugs only recently approved no such information was found. In the discussion a comparison is given with existing recommendations. It should be considered that a detailed analysis of the existing data have not performed so far.

The following was added to the Discussion:

Lines 798-805:

“This is in line with recommendations by several authors. Ethambutol and isoniazid have been classified by the American Academy of Pediatrics to be compatible with breastfeeding [110]. However, for some of the drugs monitoring of the infants for clinical signs and symptoms is recommended. For clofazimine, it is noted that discoloration of the skin may occur (LactMed®). Because of a possible influence on the gastrointestinal flora, infants whose nursing mothers receive levofloxacin should be monitored for diarrhea, thrush or diaper rash (LactMed®). Also rifampicin is classified as compatible with breastfeed, however, monitoring the infant for jaundice is recommended [110].”

Lines 806-807:

“We did not find a publication dealing with the use of pretomanid in breastfeeding women with respect to the safety of the infant.”

Lines 815-821:

“In the SPC of the European Medicines Agency, the following sentence states the regulatory view on the use of this drug in breastfeeding women, “Because of the potential for adverse reactions in breastfed infants, a decision must be made whether to
discontinue breast-feeding or to discontinue/abstain from bedaquiline therapy taking into account the
benefit of breast-feeding for the infant and the benefit of therapy for the mother (Summary of Product Characteristics, European Medicines Agency, last updated 23/11/2021).”

Lines 825-827:

“For pyrazinamide infants should be monitored for the rare side effects observed in adult patients such as clinical signs for thrombocytopenia and of liver toxicity (loss of appetite, nausea and vomiting, jaundice) LactMed®. For other anti-TB drugs no authoritative statement was found.”

7) There are some of punctuation and typographical errors throughout in the manuscript. kindly correct it.

Thank you for your comment. The manuscript was checked and corrected accordingly.

Thank you for your thorough review!

Round 2

Reviewer 5 Report

The authors have addressed all the concerns raised in the previous version of the manuscript and the quality has much improved after incorporating required modifications. Therefore, the manuscript may be considered for publication in this Journal.